# Active Learning with Crowd Sourcing Improves Information Retrieval

**Zhuotong Chen** [1]  **Yifei Ma** [2]  **Branislav Kveton** [2]  **Anoop Deoras** [2]

## Abstract

In this work, we show how to collect and use human feedback to improve complex models in information retrieval systems. Human feedback often improves model performance, yet little has been shown to combine human feedback and model tuning in an end-to-end setup with public resources. To this end, we develop a system called Crowd-Coachable Retriever (CCR),[1] where we use crowd-sourced workers and open-source software to improve information retrieval systems, by asking humans to label the best document from a short list of retrieved documents to answer a randomly chosen query at a time. We consider two unique contributions. First, our exploration space contains millions of possible documents yet we carefully select a few candidates to a given query to reduce human workload. Secondly, we use latent-variable methods to cross-validate human labels to improve their quality. We benchmark CCR on two large-scale information retrieval datasets, where we actively learn the most relevant documents using baseline models and crowd workers, without accessing the given labels from the original datasets. We show that CCR robustly improves the model performance beyond the zero-shot baselines and we discuss some key differences with active learning simulations based on holdout data.

## 1. Introduction

Recent developments in large language models (LLMs) have led to great capabilities in cognitive information retrieval (IR) and answer generation tasks (Lewis et al., 2020; Ouyang et al., 2022; Wei et al., 2022; Bubeck et al., 2023).

A surge of interests have been observed in their adaptation to applications bearing socioeconomic interests. In these applications, Lewis et al. (2020) showed that fine-tuning could lead to significant improvements. However, the fine-tuning methods were only developed for domains with labeled query-answer pairs. The problem of adaptation remains largely open for new domain containing novel knowledge or patterns, such as online retail, enterprise search, and employee match-ups, where active learning from human feedback is often required (Schütze et al., 2008; McMahan et al., 2013; Covington et al., 2016; Kang & McAuley, 2018; Choo & Siow, 2006). In those cases, human feedback can be instrumental in fine-tuning IR models. By leveraging user interactions such as clicks, dwell time, and query reformulations, the model can be trained to better understand and predict user preferences.

Specific to the IR domain, the most common feedback is human comparison of the model-retrieved candidate items. Utilizing this type of feedback poses a chicken-and-hen dilemma, where improvements in the models often depend on high-quality labels, whose collection, in turn, heavily depends on the retrieval quality of the models themselves (Settles, 2009). The need for comparative feedback, including details such as position shuffling and counterfactual reasoning, is often noted and discussed in related works concerning online feedback (Yue et al., 2012; Zoghi et al., 2016; Bottou et al., 2013; Swaminathan & Joachims, 2015; Ouyang et al., 2022; Nguyen et al., 2016; Kwiatkowski et al., 2019). In practice, however, the opportunity to run online experiments on large-scale systems is often limited due to the associated costs or risks. Therefore, it is desirable to consider an accessible alternative setup to study the key challenges in these human-in-the-loop problems.

Another challenge is the process to collect high-quality annotations. It requires careful consideration of the guidelines given to annotators, a process for handling ambiguous or difficult cases, and mechanisms to ensure the reliability and consistency of the annotations. This process is often time-consuming, expensive, and requiring expertise in both the application domain and machine learning. Existing literature on human labeling for information retrieval often relies on contracted experts passing qualification tests (Ouyang et al., 2022; Nguyen et al., 2016; Kwiatkowski et al., 2019). However, contract negotiation is often a barrier to entry by

[1]University of California, Santa Barbara; work completed during internship at AWS AI Labs. [2]AWS AI Labs, Amazon Web Services, Santa Clara, USA. Correspondence to: Yifei Ma <yifeim@amazon.com>.

Interactive Learning with Implicit Human Feedback Workshop at ICML 2023.

[1]`https://github.com/awslabs/crowd-coachable-recommendations`

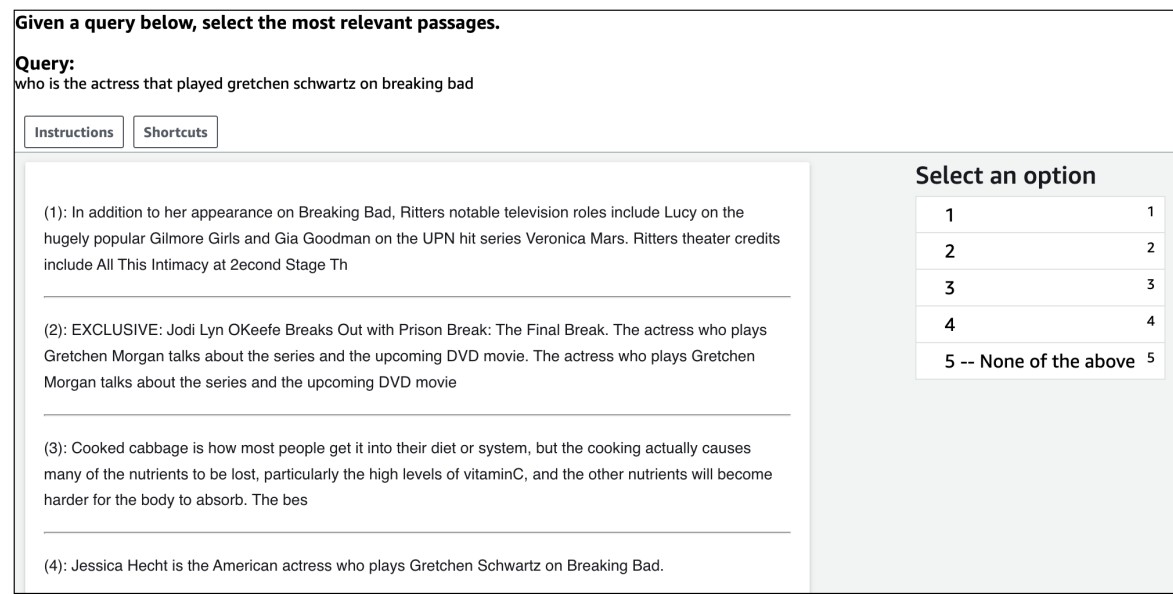

*Figure 1.* Human intelligence task on MTurk. Four candidates are displayed in random orders, each of which is truncated to at most 250 characters. A "none of the above" option is provided to reduce labeling noise. Our work shows that after collecting a few thousand labels, we can significantly improve neural retrieval models beyond zero-shot baselines according to double-blind human evaluations (Figure 3).

general researchers with relatively lower labeling budgets. More generally, crowd-sourcing systems such as Amazon Mechanical Turk allow for easy access to publicly available workers on a pay-per-label basis. The label qualities in these systems may not be guaranteed and must be managed by the requester through means of worker qualification, trick questions and/or payment rejections (Litman & Robinson, 2020; Peer et al., 2014; Buchanan & Scofield, 2018).

In this paper, we develop a new system called Crowd-Coachable Retriever (CCR) to collect human labels to improve an IR system, using only publicly available human resources and reproducible training procedures. CCR addresses the key challenges mentioned above as follows: For performance bootstrapping, CCR starts with strong zero-shot models based on popular field choices of BM25 (Robertson et al., 2009) and Contriever (Izacard et al., 2021). CCR then mixes the output candidates of BM25 with the improved dense retriever to prepare a diverse set of ranking candidates, including a "None of the Above" (N/A) option, for human comparisons; the template for the human intelligence task interface is shown in Fig. 1. To improve label quality which is crucial to the final performance, we filter the human workers based on their past history of approval ratings and further analyze their provided labels using a combination of attention checks and a latent-variable technique, famously known as Dawid-Skene (DS) method (Dawid & Skene, 1979; Sinha et al., 2018). Finally, we employ the best practices in model fine-tuning to update the model after each batch of hundreds of labeling tasks.

CCR can be used for a variety of problems, but we mostly benchmark the performance using public datasets for semantic search, based on MS-MARCO challenge (Nguyen et al., 2016) and Natural Questions (NQ) (Kwiatkowski et al., 2019). We discard the training data to craft domain-adaptation scenarios where the semantic search patterns have to be learned by active learning with comparative human feedback. We show that by using publicly accessible models, crowd-sourced workers with general-domain background knowledge, and a relatively small amount of labeled data, we can significantly improve the model performance beyond the initial zero-shot baselines, further matching or outperforming our simulations of active learning based on the discarded ground-truth labels under equal labeling budgets. CCR greatly simplifies the requirements for models and labelers, allowing easy improvements and potential extensions of IR systems for new applications.

## 2. Active Learning with Human in the Loop

This section provides guidelines for using human labels to improve a search system in a new domain. To begin with, we establish strong baselines from unsupervised retrieval models, to gain trust with humans to obtain high-quality labels (Section 2.2). Then, we design active learning systems with a focus on practical approaches to diverse challenges (Section 2.3). When working with human labelers, we find quality control to be a key component and adopt Dawid & Skene (1979) algorithm to suppress noisy labels

**Algorithm 1** Crowd-Coachable Retriever

---

**input** A set of queries $q_i \in \mathcal{Q}$, all documents $d_i \in \mathcal{D}$
**output** Fine-tuned neural retriever $M_k$ by human labels.

1: Construct $M_0$ and baselines $M_{\text{bm25}}, M_{\text{rand}}$ (Sec. 2.2).
2: **for** step $s = 0, \ldots, k-1$ **do**
3:     Retrieve candidate answers by ensemble models $M_s$, $M_{\text{bm25}}$, $M_{\text{rand}}$ for the next batch of queries as $\{(q_i, d_{i_1}, d_{i_2}, \ldots, d_{i_k})\}_{i=1}^{|\mathcal{Q}|/k}$ (Sec. 2.3).
4:     Collect labels (Sec. 2.4) and apply DS algorithm to find $\{(q_i, d_{i_{(1)}}, d_{i_{(2)}}, \ldots, d_{i_{(k)}})\}_{i=1}^{|\mathcal{Q}|/k}$, where $d_{i_{(1)}}$ is the most preferred label (Sec. 2.5).
5:     Fine-tune $M_{s+1}$ with the labeled data (Sec. 2.6).
6: **end for**

---

(Sections 2.4 and 2.5). Finally, we use robust fine-tuning techniques to ensure improvements in each iteration (Section 2.6). Algorithm 1 shows the key steps and notations.

## 2.1. Backgrounds on Bi-Encoder Retrieval Models

Throughout the paper, we use bi-encoder similarity models for information retrieval. These models encode a query $q_i$ and a document $d_i$ using a shared network $f_\theta(\cdot)$, parametrized by $\theta$, followed by dot-product to produce relevancy scores,

$$u(q_i, d_i, \theta) = f_\theta(q_i)^\top f_\theta(d_i). \tag{1}$$

In the initial zero-shot learning stage, documents and queries are presented independently with an unsupervised learning objective. In later stages, training data is presented in comparison groups $(q_i, d_{i_{(1)}}, d_{i_{(2)}}, \ldots, d_{i_{(k)}})$, where one selected document, $d_{i_{(1)}}$, is contrasted against all other documents $\{d_{i_{(j)}} : \forall j > 1\}$ dismissed by the human annotator at the time. Both objectives are used to train $\theta$ and our goal is to show good initial results as well as continuous improvements as we collect more comparison labels of the latter kind. Performance evaluation, as well as retrieval-based active learning which we discuss in Section 2.3, is based on the top $k$ answers from all documents:

$$(d_{i_1}, \ldots, d_{i_k} | q_i) = \text{top}_k \{u(q_i, d, \theta) : d \in \mathcal{D}\}. \tag{2}$$

## 2.2. Unsupervised Learning for Zero-Shot Retrieval

In the proposed active learning framework, we employ a zero-shot retrieval model as the initial step. This model can be categorized into two types in the realm of information retrieval: the dense retrievers, represented by Contriever (Izacard et al., 2021), and the sparse retrievers, represented by BM25 (Robertson et al., 2009). Aiming to create diversity in the labeling tasks, we combine the results yielded by both these models. Our ultimate goal is the construction of a retriever, fine-tuned by human feedback, that exceeds the performance of both the aforementioned models.

**BM25** (Robertson et al., 2009) is a strategy underscored by its simplicity and effectiveness, and operates on unique search keywords. Its principle is the retrieval of documents based on keyword matches, wherein each match is additionally weighed by an inverse-document frequency (IDF) term. This IDF term indicates the rarity and thus the uniqueness of information the keyword provides, along with other factors concerning document lengths. This principle gives BM25 robust empirical results on large retrieval datasets. Despite its strengths, BM25 exhibits limitations in understanding semantic or complex logic. Moreover, it lacks learnability, even if we collect labels about the desired patterns.

**Contriever** (Izacard et al., 2021) is a neural embedding model that learns to embed each document into a fixed-dimensional vector. It adheres to the bi-encoder structure, where it forms a positive pair from two random views of the same document (through token masking) and multiple negative pairs from unrelated documents in the same mini-batch. The loss function is:

$$\min_\theta \frac{1}{N} \sum_{i=1}^{N} - \frac{\exp(u(d_i^{(1)}, d_i^{(2)}, \theta)/\tau)}{\sum_{j=1, j\neq i}^{N} \exp(u(d_i^{(1)}, d_j^{(2)}, \theta)/\tau)}, \tag{3}$$

where $\tau$ is a temperature parameter and $(d^{(1)}, d^{(2)})$ are the two random views of the same document. Notice that the loss function does not depend on labeled query-answer pairs and thus Contriever can be classified as a zero-shot model.

## 2.3. Acquisition Strategy for Active Learning

CCR adopts an active learning paradigm by designing multiple-choice questions based on the initial zero-shot models, thereby enabling learning from human labels. This involves a strategy based on random queries and a greedy selection of top-ranked answers sourced from an ensemble of models. Specifically, we select the two highest ranked answers from the current iteration of the CCR model, as denoted by (2), along with one top answer from the BM25 model, and one randomly selected answer as an attention check. Duplicate selections are removed by opting for subsequent candidates from the respective models. The inclusion of the random answer yields ground-truth signals pertaining to labeler quality, which we subsequently leverage to calibrate a more sophisticated labeler evaluation algorithm, Dawid-Skene (DS), as detailed in Section 2.5.

Our main concern in the ensemble design is to break homogeneity in the candidate options so that we obtain a correct label if there is one. This consideration arises due to the potential for crowd workers to contribute noisy labels, which, in the presence of ambiguous ground-truth labels, can impede the efficacy of the DS algorithm in evaluating labelers and refining labels. Furthermore, our ensemble design exhibits links to optimal designs intended for information gain,

although the analysis of deep neural networks' properties can be challenging. Typical alternatives include Thompson sampling (Chapelle & Li, 2011), upper confidence bounds (Li et al., 2010; Zhou et al., 2020), or disagreement regions (Foster et al., 2018). There has also been research focusing on conservative bandits or trust-region policies which respect psychological factors, requiring the active learner to establish trust with its users by eschewing risky explorations (Wu et al., 2016; 2017; Zhu & Kveton, 2022). Finally, Bastani et al. (2021) demonstrated that in certain contexts similar to ours, where part of the contexts (the queries) are randomized, greedy selection of the candidate answers can be optimal. Thus we keep randomized queries for simplicity.

## 2.4. Human Annotations

Despite the challenges encountered in the end-to-end execution of active learning methods, one of the most significant novelties, and concurrently challenges, of our study has been the use of human annotations. Through the course of the research, we iteratively enhanced our label collection mechanisms, with each improvement directly influencing the performance of our model. We detail our final methodological approach herein, as employed in our experiments.

Our main form of data collection leverages multiple-choice questions, as depicted in Figure 1. We categorize the answers selected by humans as positive, while non-selected responses are considered hard negatives. We further provide the annotators with a "None of the Above" (N/A) option to minimize the noise in the training dataset. Subsequent fine-tuning tasks exclude these data points, as we observe that similar examples can often be compensated by other queries where true answers are incidentally included. For instance, Figure 4 in Section 3.4 illustrates how CCR ultimately learns to retrieve documents containing actual numbers in response to a user query for bank routing numbers, despite initially failing to do so.

Given our reliance on crowd-sourcing systems for label collection, we lack prior control over labeler quality. To counter this, we impose filters based on the labelers' past performance, favoring US-based "master" labelers who have completed a large number of labeling tasks while maintaining high payment approval rates. Subsequently, we distribute each question to multiple (three) labelers, with the order of the candidates randomly altered, and utilize the DS algorithm to identify the most probable answers. The DS algorithm evaluates the labelers based on their mutual agreement frequency. We found the DS ratings to be reliable, as evidenced by the frequency with which a labeler fails our attention checks based on the random answers.

## 2.5. Dawid-Skene Voting

Dawid & Skene (1979), shortened as DS, is a latent-variable

model that improves over simple majority voting by jointly modeling the labelers' confusion matrices among different classes. It is an essential component for our data cleaning and labeler rewarding system. We define some variables only in this subsection to conform to common conventions. Let $i \in I, j \in J, k \in K$ be the indices for a task, a labeler, and a class, respectively. Let

$$\boldsymbol{\Theta}_{j::} = \sigma(\gamma_j)\mathbf{I}_K + \frac{\sigma(-\gamma_j)}{K}\mathbf{1}_K\mathbf{1}_K^\top \tag{4}$$

be the confusion matrix with a simplified signal-to-noise parameter $\gamma_j$ for each labeler $j$, where $\mathbf{I}_K$ is the identity matrix for producing correct labels, $\frac{1}{K}\mathbf{1}_K\mathbf{1}_K^\top$ is for a uniform confusion distribution, and $\sigma(\gamma) = \frac{e^\gamma}{1+e^\gamma}$ is the sigmoid activation function. DS assumes that each labeler labels each task independently. For task $i$ with collected labels $D_i = \{(i_n, j_n, y_n) : i_n = i\}$, we can write out the complete likelihood function assuming that the true label is $z_i$, as

$$P_{\boldsymbol{\gamma}}(z_i, D_i) = P(z_i)P_{\boldsymbol{\gamma}}(D_i|z_i) = \frac{1}{K}\prod_{n:i_n=i}\boldsymbol{\Theta}_{j_n z_i y_n},$$

where the multiplication over $\{n : i_n = i\}$ aggregates over all labels collected independently from labelers on the same task and we assume uniform class-prior distribution $P(z_i) = \frac{1}{K}$. We then write out the learning objective:

$$\max_{\boldsymbol{\gamma}} \sum_{i=1}^{I} \log(P_{\boldsymbol{\gamma}}(D_i)) = \sum_{i=1}^{I} \log\left(\sum_{z_i=1}^{K} P_{\boldsymbol{\gamma}}(z_i, D_i)\right) \tag{5}$$

Since the objective contains a latent variable, we optimize it by expectation-maximization, by alternating between:

$$\text{E-step: } q(z_i) = P_{\boldsymbol{\gamma}}(z_i|D_i) = \frac{\prod_{n:i_n=i}\boldsymbol{\Theta}_{j_n z_i y_n}}{\sum_{z'=1}^{K}\prod_{n:i_n=i}\boldsymbol{\Theta}_{j_n z' y_n}}$$

$$\text{M-step: } \max_{\boldsymbol{\gamma}} \sum_{i=1}^{I}\left(\sum_{z_i=1}^{K} q(z_i)\log(P_{\boldsymbol{\gamma}}(z_i, D_i))\right), \tag{6}$$

where the M-step can be implemented by automated gradient descent in a neural network module.

Notice that DS assumes independent observations, so we shuffle the positions of our answers to avoid collective biases due to the orders in the displays. However, we cannot shuffle the position of the N/A class and thus we exclude the N/A class during training for the labelers' quality parameters $\gamma_j$. We then include the N/A class in a final run of the E-step to infer the true labels. Finally, to protect labelers with small contributions from receiving extreme scores, we use weight decay on $\gamma_j$, which produces a similar effect to hierarchical Bayesian priors.

A low $\gamma_j$ score indicates that the labeler is repeatedly inconsistent with the rest of the labelers. There are many reasons

for the inconsistency, some of which can be attributed to unclear query statements or lack of diversity in answer options, we observe that extremely low $\gamma_j$ scores ($< 0.15$) often indicate noisy labelers based on our ground-truth randomness signals, shown in Figure 5 in Section 3.5. In these cases, we block the labeler from future tasks.

### 2.6. Supervised Fine-Tuning

The last step of the proposed CCR is model fine-tuning. We use *Multiple Negative Ranking Loss (MNRL)* (Henderson et al., 2017) to fine-tune the retrieval model. Based on the cleaned labels from the data collection step, we receive a batch of $b$ training data groups $\{(q_i, d_{i(1)}, d_{i(2)}, \ldots, d_{i(K)})\}_{i=1}^b$, where $d_{i(1)}$ is the voted positive answer and $d_{i(k)}$ are the remaining dismissed answers for query $q_i$. The objective function of MNRL is defined as follows,

$$\min_{\boldsymbol{\theta}} \frac{1}{b} \sum_{i=1}^b - \log \frac{\exp\left(u(q_i, d_{i(1)}, \boldsymbol{\theta})/\tau\right)}{\sum_{j=1}^b \sum_{k=1}^K \exp\left(u(q_i, d_{j(k)}, \boldsymbol{\theta})/\tau\right)}, \tag{7}$$

where $\tau$ is a temperature parameter set to $0.05$. MNRL is similar to the cross-entropy loss for multiple-choice questions, traditionally cast as $- \log \frac{\exp\left(u(q_i, d_{i(1)}, \boldsymbol{\theta})/\tau\right)}{\sum_{k=1}^K \exp\left(u(q_i, d_{i(k)}, \boldsymbol{\theta})/\tau\right)}$, but it extends the negative examples to all documents in the mini-batch, including documents retrieved from same queries and other queries. In this way, MNRL also resembles the loss function of the zero-shot model (3), where including the weak negative examples effectively regularizes the solution by the prior objective.

## 3. Experiments

In this section, we document our active learning experiments using MS-MARCO and NQ datasets as two examples. We start with state-of-the-art unsupervised retrieval models but we aim to make real progress beyond our complex NLP baselines by learning from human labelers. This is challenging because we make choices for many subcomponents, namely candidate retriever, labeler noise control, and model fine-tuning with only indirect or delayed feedback; and we only have budgets for a limited number of experiments. To justify the use of human labelers, we make the assumption that the target domain contains different properties than the training domain and the difference can only be demonstrated by human labelers. In fact, the datasets we chose contain semantic search patterns that are not learnable by lexical search methods and, to an extent, unsupervised deep models without search-specific labels. We show empirical results (Sections 3.2 and 3.3), visualization results (Section 3.4), and additional details at the end of the section.

### 3.1. Setups

**Pre-Training:** We used the Contriever model borrowed directly from huggingface, which is itself a pre-trained model using Wikipedia and CCNet (Wenzek et al., 2020). The Contriever model has not seen question-answer pairs in any search domains, but it can be improved by learning from human-comparison feedback. We also used BM25 to provide another sparse retriever baseline. We used BM25 with system-wide default $k_1 = 0.9$ and $b = 0.4$.

**Active Learning:** We used the dev set for labeling, which includes 6980 queries and 8 million candidate answer passages in MS-MARCO experiments and 3452 queries and 3 million candidate answer passages in NQ experiments. We split the dev set into four mini-batches where each batch contains 1745 and 850 queries, respectively for the two datasets. For each mini-batch, we applied the active learning designs outlined in Section 2.3, followed by label-quality controls (Sections 2.4 and 2.5) and model fine-tuning (Section 2.6).

**Evaluations:** Evaluations for all our experiments include generalization test accuracy (based on ground-truth labels for queries not present in the collection phase), and human evaluations (i.e., not using the ground-truth labels provided in the original datasets). The evaluation metrics are common in the search and recommendation community:

- Mean Reciprocal Ranking (MRR@100) is a position-aware metric, defined as the inverse of the position of the first correct answer in the top-100 retrievals. MRR@100 is a common metric to measure retrieval model performance (Nguyen et al., 2016);
- Recall@4 is a position-insensitive metric, defined as the number of correct answers in the top 4 retrievals divided by the total number of ground-truth solutions (capped at 4 for fair comparisons). We consider Recall@4 since 4 answers are presented to human annotators.

### 3.2. Generalization Performance

To show generalization, we held out the last batch in the dev set and used the models learned from previous batches of data collections to retrieve passages in the last batch of queries. This corresponds to 1730 test queries on MS-MARCO and 857 test queries on NQ for holdout evaluation.

Figure 2 shows our results for active learning with human (and oracle) labelers. The curves reflect the progress of active learning, starting with the zero-shot Contriever model, which is a strong baseline model as the initial recall rates with the top-4 retrievals were around 20% and 31%, respectively, and the MRR scores were also high. With more examples collected from human-comparison feedback, the CCR performance improved significantly, outperforming both the initial Contriever model as well as the BM25 model.

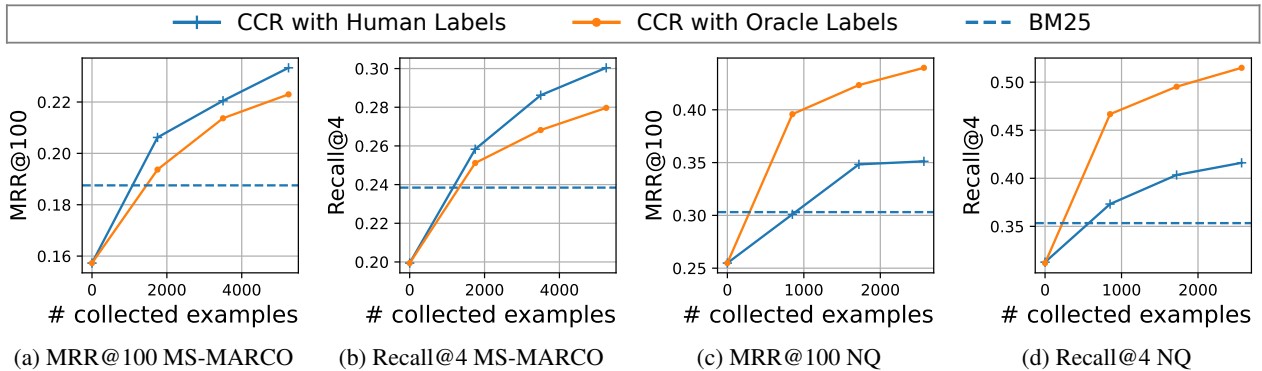

(a) MRR@100 MS-MARCO    (b) Recall@4 MS-MARCO    (c) MRR@100 NQ    (d) Recall@4 NQ

*Figure 2.* Generalization performance using the last batch of queries as hold-out. Human-labeled models outperformed zero-shot model baseline and BM25 baseline. We also include oracle-labeled models for completeness, despite that oracle labels are not feasible in on-domain learning problems.

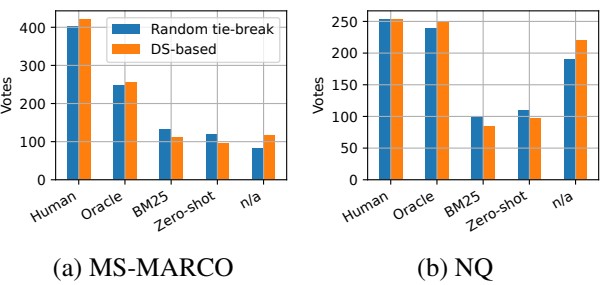

(a) MS-MARCO          (b) NQ

*Figure 3.* Human preference between model fine-tuned with human labels, model fine-tuned with oracle models, BM25 baseline, and zero-shot baseline. Results aggregated from 1000 unique queries.

We believe that having clean labels through DS voting contributed to better generalization. Our earlier experiments did not include DS voting and our preliminary results showed negative comparisons with the BM25 baselines.

We also include CCR with oracle labels for completeness, where the fine-tuned model also improved generalization performance compared with BM25 and zero-shot baselines. The simulation oracles returned the ground-truth answers if they were included in the four candidate answers in the labeling task and N/A if they were missed out. Notice that the simulation oracles present biases in the evaluation. This is because the simulation labels are collected based on Bing and Google search results, which allows the oracle-guided models to pick up any biases from the search engines and use them to their favor during hold-out tests. Besides, the simulation oracles always yield clean labels, which removes one challenge in the crowd-sourcing process. A fairer comparison should include human evaluation, which we describe next.

### 3.3. Human Evaluations

We consider human evaluation of the learned models. We used queries from the dev set with one candidate answer from each of the four different models: CCR with human labels, CCR with Oracle labels, BM25, and the initial zero-shot model. To reduce noise, we picked queries where the four models generated four distinct answers, without explicitly biasing towards any models. We downsampled the evaluation set to contain no more than 1000 queries.

We collected three independent labels per query and used majority voting and DS voting to aggregate the results, shown in Figure 3. On the MS-MARCO dataset, CCR with human labels is a clear winner and the initial zero-shot models are clear losers. On the NQ dataset, CCR with human labels outperformed the two zero-shot baselines, as we expect. On both datasets, we also observe that CCR with human labels outperformed CCR with oracle labels. This shows that we can obtain the desired results using simple and pure setups without relying on simulation oracles whose construction relies on cleaner labels from superior models like Bing or Google search engines.

### 3.4. Knowledge Transfer Visualization

To support our generalization claim, we can also visualize which label $(q_0, d_0)$ enables the fine-tuned model to retrieve the correct answer for an unseen query $(q_*, d_*)$. This is done by finding positive changes to the utility function for the target pattern, $u(q_*, d_*, \boldsymbol{\theta}_*)$, defined in (1). Usually, explainability requires one to consider the complex relations of all training examples. Here, we simplify the approach with only marginal changes from a one-step gradient of model parameters:

$$\boldsymbol{\theta}_* = \boldsymbol{\theta}_0 + \eta \partial_{\boldsymbol{\theta}}(u(q_0, d_0, \boldsymbol{\theta}))|_{\boldsymbol{\theta}_0},$$

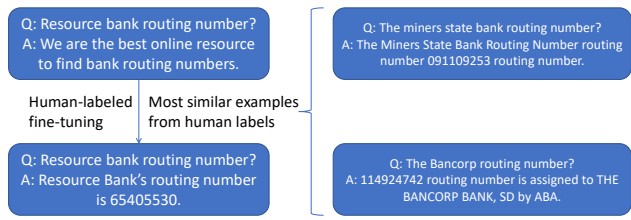

Figure 4. An example for knowledge transfer visualization

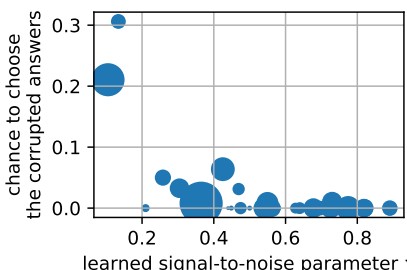

Figure 5. Dawid-Skene calibration against ground-truth labeler noise, collected from Iteration 2 in NQ experiments. Each dot represents a labeler and its size shows the number of labels they provided. A high correlation can be seen between the two methods.

where $\eta$ is the step size, $\boldsymbol{\theta}_0$ and $\boldsymbol{\theta}_*$ represent the model parameters before and after the gradient update, respectively. By inserting the update to the prediction of the target pattern, we see the relative change in the utility score,

$$
\begin{aligned}
&u(q_*, d_*, \boldsymbol{\theta}_*) - u(q_*, d_*, \boldsymbol{\theta}_0) \\
&= u(q_*, d_*, \boldsymbol{\theta}_0 + \eta \partial_{\boldsymbol{\theta}}(u(q_0, d_0, \boldsymbol{\theta}))|_{\boldsymbol{\theta}_0}) - u(q_*, d_*, \boldsymbol{\theta}_0), \\
&\approx \eta \partial_{\boldsymbol{\theta}}(u(q_*, d_*, \boldsymbol{\theta})|_{\boldsymbol{\theta}_0})^T \partial_{\boldsymbol{\theta}}(u(q_0, d_0, \boldsymbol{\theta})|_{\boldsymbol{\theta}_0}), \quad (8)
\end{aligned}
$$

where the last step follows from gradient definition, $u(x_0 + \Delta x) - u(x_0) \approx \partial_x u(x_0)^\top \Delta x$. Therefore, if $(q_0, d_0)$ is similar to $(q_*, d_*)$, we may observe high dot-product scores of their gradients, which indicates a positive gain to the utility function in the approximation.

In Figure 4, we show a query "the resource bank routing number", which has two components: "resource bank" and "routing number". The zero-shot model failed to recognize the correct meaning of these objectives. It favored answers that contain exact matches of the entity instead of answers that explain the query. In this case, it incorrectly identified "resource" as assets. When the zero-shot model was fine-tuned on a training dataset that does not contain this specific query, it returned the correct answer. By performing the gradient similarity search in Eq. (8) on all training samples, we identified two query-answer pairs that contributed the most to the utility score of the target query-answer pair. In Figure 4, both relevant queries have the same structure as the target query, "The bank routing number", which provides semantic information to the zero-shot model to generalize the knowledge on "Resource bank routing number".

### 3.5. Dawid-Skene Voting and Rating

Figure 5 shows the DS evaluation score and the labeler's chance to choose the corrupted answers, which can be up to 0.2 if the labeler chooses completely random answers. (A corruption score that is greater than 0.2 suggests that the labeler may be confused and this was an uncommon situation, especially after we improved the instructions in later steps.) We see a high correlation between the two, showing that the DS score is a reliable way to assess labeler qualities. We set a threshold of $\gamma = 0.15$ as our payment rejection criterion.

The rejection rate in Figure 5 was 13.2% by the number of labels. Our highest rejection rate was 27% in our initial iteration, yet we attribute that to our lack of reminders in the instructions and we reconciled the labelers who reach out to us. Our lowest rejection rate was 0%, yet this can be a rare phenomenon. The rejection rate is random in nature and it can highly depend on the time of day and other external factors. The rejected answers were re-released to gather clearer signals, though we did not find significant changes in fine-tuning performance. Since we work with very few labels, we find that bad labels can have significant undesirable effects on the performance. DS led to significant improvements over majority voting.

### 3.6. Additional Results

Appendix A provides additional details for our human task designs as well as our prior experience with other crowd sourcing systems, Appendix B provides experimental results on on-domain active learning, which is a more traditional active learning setting that we omit due to space constraints, Appendix C shows examples for each winning label in our human evaluation experiments, including the N/A label.

## 4. Related Work

We briefly cluster related work into the following topics.

**Human Labeling**   Human labeling has fueled machine learning to achieve many important milestones in computer vision (Russakovsky et al., 2015), semantic search (Nguyen et al., 2016; Kwiatkowski et al., 2019), and most recently, instruction following (Ouyang et al., 2022). A list of labeling work specific to reading comprehension, logic reasoning, dialogue systems, and other natural language applications can be found in the references in (Ouyang et al., 2022). Besides these fundamental labeling efforts, more common are expert annotations for specific applications, such as product tagging (Inoue et al., 2017), autonomous driving (Huang et al., 2018), science (Krallinger et al., 2015), health (Pinsky &

Dubrawski, 2014), and law enforcement (Dubrawski et al., 2015). This is a very short list of possible applications with human annotations, presented for inspirational purposes.

**Online Exploration** Our work is related to online exploration as we share the same type of feedback for information retrieval systems. While many techniques have been developed under the theoretical promise of long-term model improvements (Li et al., 2010; Chapelle & Li, 2011; Foster et al., 2018; Riquelme et al., 2018; Schulman et al., 2015), exploration in practice is often much more myopic, focusing on heuristics such as rule-based promotions or topic-wise calibrations (Steck, 2018). Our work shows model improvements in a completely isolated setting, validating the long-term effects. Along this line, we share similarities with works that promote unbiased learning from randomized recommendations (Bottou et al., 2013; Swaminathan & Joachims, 2015; Zoghi et al., 2016; Wang et al., 2017). Libraries for online exploration exists (Joachims et al., 2018; Bietti et al., 2021; Foster & Rakhlin, 2020) and the form of comparative feedback is analyzed under the framework of dueling bandits (Yue et al., 2012; Sui et al., 2018). A closely related field is active partial labeling (Hu et al., 2019; Durand et al., 2019; Yun et al., 2021).

**Crowdsourcing in Active Learning** The domain of active learning utilizing crowdsourced contributors has a long history (Li et al., 2016). In Pfeiffer et al. (2012), the query is a pair of images with dots and the crowdsourced workers are asked to identify the image with the most dots. In Chen et al. (2013), the query is a pair of documents, and the crowdsourced workers are asked to rank them by reading difficulty. In both works, the queries were adaptively selected. Our work differentiates itself primarily in terms of the domain, the complexity of the neural network models employed, and the difficulty of the tasks, which are typically handled by trained labelers.

**Human Feedback on Language Modeling** Pre-trained language models such as GPT-3 (Radford et al., 2019) and Facebook's XLM and XLM-R (Conneau et al., 2019) have significantly advanced the field of natural language understanding. However, it has been observed that the behavior of these large language models does not always align precisely with user intentions (Ouyang et al., 2022). This discrepancy can be attributed to both biased training objectives and data. For instance, a contrastive loss may lead to popularity bias, where frequently occurring answers deviate from users' actual preferences (Zhang et al., 2022). Furthermore, training data in the language domain, often constructed with assistance from other search engines (Nguyen et al., 2016), can introduce undesired bias into the training process. Studies have shown that aligning the output of a trained language model with human behavior can be achieved via fine-tuning

with human feedback (Ouyang et al., 2022). Notably, recent work on ChatGPT (Ouyang et al., 2022) and related topics have underscored the considerable potential of human-in-the-loop machine learning. One way to appreciate the success of ChatGPT is to understand it as a pre-fine-tuned model in common domains of interest. In the specific domains related to search and recommendation, it is still relatively unclear how to fine-tune the models based on human feedback. For example, Gao et al. (2022; 2023); Zhang et al. (2021) documented various attempts at using pre-trained language models for direct movie recommendation. While these efforts provide a solid starting point, we posit that incorporating human feedback can further enhance domain adaptivity, a topic we aim to explore in this paper.

## 5. Conclusions

In this work, we present CCR, an end-to-end active learning framework designed to improve information retrieval models based on crowd-sourced human feedback. Keys to our proposal are the use of pre-trained retrieval models, comparative feedback from diverse candidates, and mechanisms to inspire human labelers to provide high-quality labels. Empirical results on representative real-world datasets showcase the success of our proposal. While our work is a first step at combining active learning, crowd sourcing, and information retrieval, we observe clear opportunities for improvements. In our framework, retrieving diverse candidates is not only important for model improvements, but also crucial for the human labelers to perform their ranking duties and the learners to assess the label qualities with strong confidence. We have yet to establish a principled metric for diversity from human perspectives. Besides the diversity challenge, labelers may also lack consensus due to the difficulty of the tasks. While some tasks have been labeled with the N/A class, it may be desirable to systematically screen all tasks for difficulty and place the more difficult tasks into a learning curriculum (Bengio et al., 2009; Graves et al., 2017). Moreover, our current approach is limited to comparing answers given a question. An equally common approach in natural question answering is to label the questions given the answers. This approach resembles Bayesian posterior inference and it implicitly increases the diversity of the labeled examples. Along this line, (Rajpurkar et al., 2016) produced the well-known SQuAD dataset, (Ravichandran et al., 2019) showed improvements on few-shot tuning. We choose the current labeling regime to stay consistent with online experiments in recommendation and search systems. Finally, our CCR approach is agnostic to model architecture and we can follow the diversity approach outlined in (Lewis et al., 2020) for generative modeling. Despite the huge success of instruction-following LLMs, their adaptation in custom domains still requires additional human labels, especially if the performance or risk is of great importance.

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

# A. Details on Human Annotations

In this work, we used Amazon Mechanical Turk (MTurk) to collect crowd-sourced feedback. Fig. 1 shows our human intelligence task (HIT) design. In addition, we provided a short instruction along each task as shown in Fig. 6. Each task requires a human annotator to choose the most relevant passage from four possible candidates. Since those candidates are extracted as the top-4 answers from our model, there are no guarantees that the true answer is contained in those four candidates. Therefore, we allowed the human annotator to select "None of the above" if no relevant passages were present.

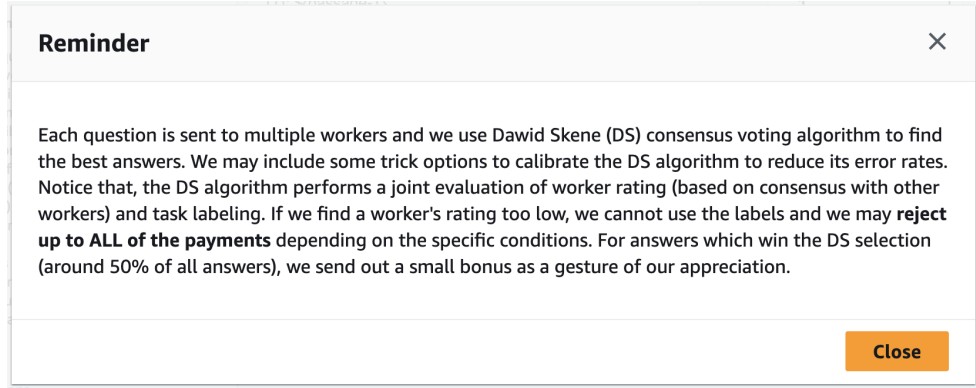

*Figure 6.* Instruction and reminder in Human intelligence task design on MTurk.

An important aspect of HIT task is the design of pay structures and instructions. Figure 6 shows an example of desired labels as well as additional reminders that we may disapprove payments as a mechanism to control for label qualities. We paid 0.45 USD for 3 annotators per task and used the following criteria for workers to ensure high quality: *Location: US; Minimal approval rate: 95%; Number of HITs approved: 1000.* One of our Turkers suggested to up the approval rate to 99% and increase the approval number to 10k while removing the "masters" badge for qualifications. We include this suggestion for future work.

We considered other environments as well, but we chose MTurk because it allows us to have direct communications with the label workers. By default, MTurk delays the payments by 72 hours. For workers with good approval histories, the uncertainty of payment rejection plays an important role and it already led to quality improvements compared with our prior experiments in other environments where payment rejections were not possible. However, unconditional approval, even if delayed, will eventually lead to deteriorating performance over time as the workers keep track of their completed tasks. We further implemented latent-variable rating systems (Section 2.4) and added clear instructions (Figure 6), which allowed us to directly measure and provide guarantees for label qualities.

**Question:** 3 levels of government in canada and their responsibilities

*Please pick correct answer:*

(1) There are three levels of government in Canada-Federal, Provincial and Municipal-each with its own set of responsibilities. Click on the image icon for more information.Try the Levels of Government game !here are three levels of government in Canada-Federal, Provincial and Municipal-each with its own set of responsibilities. Click on the image icon for more information. Try the Levels of Government game !

(2) This was not true of the Industrial Workers of the World and as a result many of its members were first and second generation immigrants. Several immigrants such as Mary 'Mother' Jones, Hubert Harrison, Carlo Tresca, Arturo Giovannitti and Joe Haaglund Hill became leaders of the organization.

(3) Hi Meagan, Yes, the Blood Sugar Solution 10-Day Detox diet is filled with nutritious whole foods including lean protein, vegetables, fruit, and healthy fats from nuts, seeds, olive oil, and coconut oil. The diet excludes foods that are common allergen and is safe for you and your baby while you are breastfeeding.

(4) There was another side to Andrew Jackson. He was a man who engaged in duels, killing Charles Dickinson in 1806. During in the First Seminole War he inflicted harsh discipline on his troops, including executions for mutiny. The necessity of some were questioned.

(5) None of the above

*Figure 7.* Human intelligence task design on Amazon Ground Truth.

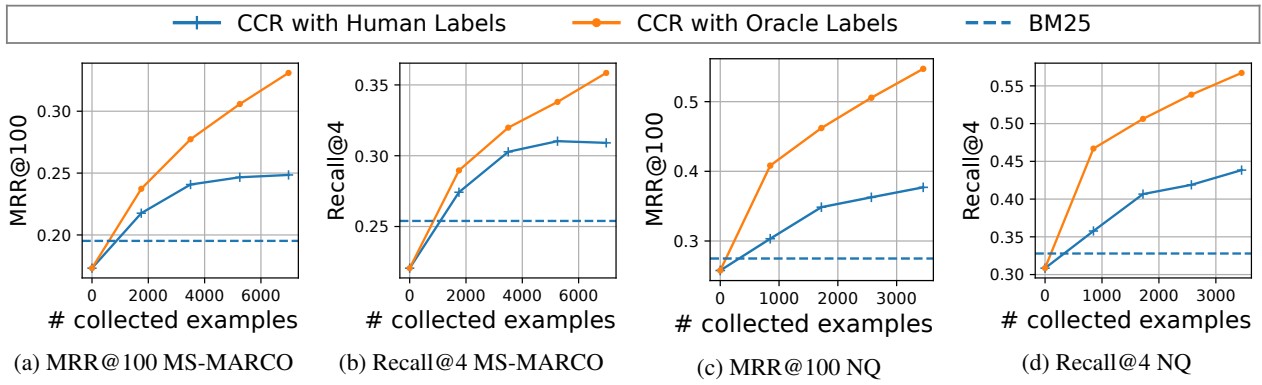

(a) MRR@100 MS-MARCO      (b) Recall@4 MS-MARCO      (c) MRR@100 NQ      (d) Recall@4 NQ

*Figure 8.* On-domain test performances based on ground-truth labels in the original dev sets. The human-labeled models showed consistent improvements despite the fact that the labelers had no prior knowledge of the ground-truth labels, showing consistency across different sources of labels. Beside the human-labeled models, we include models trained with oracle labelers, which reveal the true answers if they are included in the active-learning candidate set. Oracle-labeled models are not our focus, but they share the same infrastructure and they can validate the fine-tuning part of CCR.

## B. On-Domain Active Learning

On-domain active learning is a traditional setup for active learning. Compared with generalization tests, the on-domain setup is conceptually easier because it mixes the training queries (used for label collection) and the testing queries. However, on-domain active learning is still nontrivial and it is not to be confused with training loss in supervised learning. This is because the true documents for the known queries are still unknown, unless they are actively acquired through the multiple-choice questions which are carefully designed with model-retrieved candidate documents. Further, we collect labels from crowd workers, which are often different from the ground-truth labels collected from experts in the original datasets. On-domain active learning is both a practical setup for many real-world problems, as well as a challenge for generalization, especially across different labeling environments.

Figure 8 shows our results for active learning with human (and oracle) labelers. The patterns follow closely with Figure 2 in that CCR starts at strong zero-shot baselines and further improves by collecting and learning from human labels, surpassing the other BM25 baseline that we also considered. The relative improvements for CCR with human labels are also similar to the generalization performance in Figure 2.

Notice that the results for CCR with oracle labels are overly optimistic in Figure 8. This is because for on-domain active learning, the learners may simply memorize the correct answers exactly and reapply them at test time. The bias is even

larger when the ground-truth labels are all collected with – and therefore filtered by – Google or Bing search engines. Comparatively, CCR with human labels faithfully revealed the amount of general patterns being learned and transferred from different labeling environments.

## C. Examples from Human Evaluation Tasks

Tables C1 to C4 show examples from our final human evaluation tasks to provide more insights into the challenges of active learning in the real world. We include cases for each winning method, where our proposed CCR with human labels had the highest winning rates, shown in Figure 3 in Section 3.3. For cases where human selected "none of the above" options, we also included answers from the original datasets. Notice that CCR with human labels never had access to these labels and CCR with oracle labels only had access if these answer passages were retrieved in one of the active learning steps. Therefore the retrieval of the optimal answers is fundamentally challenging. Our work shows that we can use pretrained models with a modest budget (equivalent to 2% of what was used to collect the original labeled datasets) to demonstrate effectiveness in active learning with human feedback for cold-start search problems.

| Query | what are the 6 parts of the brain |
|---|---|
| **Human** | There are 8 parts of the brain. Heres a list of what they are and what they do. The frontal lobe is one of the 4 major divisions of the cerebral cortex. This part of the brain regulates decision making, problem solving, control of purposeful behavior |
| Oracle | What are the parts of the brain? The main parts of the brain are the frontal lobe, temporal lobe, occipital lobe, parietal lobe, hypothalamus, cerebrum, brain stem ,and cerebellum. In addition, the brain contains the corpus callosum, pituitary gla |
| BM25 | Here are some examples of functions that the brain controls: The brain is like a busy city. Each part has different functions and is made up of different types of cells. To work, different parts of the brain need to send messages to each other, and t |
| Zero-Shot | What Are the Parts of the Brain? Every second of every day the brain is collecting and sending out signals from and to the parts of your body. It keeps everything working even when we are sleeping at night. Here you can take a quick tour of this amaz |

| Query | synonym for the word evaluate |
|---|---|
| **Human** | Synonyms for evaluate in the sense of this definition. (evaluate is a kind of ...) use or exercise the mind or ones power of reason in order to make inferences, decisions, or arrive at a solution or judgments. |
| Oracle | Definition 1: evaluate or estimate the nature, quality, ability, extent, or significance of [verb of cognition] Samples where evaluate or its synonyms are used according to this definition. Synonyms for evaluate in the sense of this definition. (eva |
| BM25 | Definitions and Synonyms of evaluate Another word for evaluate What is evaluate? Definition 1: evaluate or estimate the nature, quality, ability, extent, or significance of [verb of cognition] Samples where evaluate or its synonyms are used accord |
| Zero-Shot | Here are all the possible meanings and translations of the word evaluate. Princetons WordNet(5.00 1 vote)Rate this definition: measure, evaluate, valuate, assess, appraise, value(verb) evaluate or estimate the nature, quality, ability, extent, or si |

| Query | what is a straddle |
|---|---|
| Human | Straddle(verb) to place one leg on one side and the other on the other side of; to stand or sit astride of; as, to straddle a fence or a horse. Straddle(noun) the act of standing, sitting, or walking, with the feet far apart. Straddle(noun) |
| **Oracle** | A straddle is any set of offsetting positions on personal property. One example, is a put and call option on the same number of shares of a particular security, with the same exercise price and expiration date. |
| BM25 | straddle my rail; Straddle Piss; straddle puss; straddler; Straddle Racking; Straddlers Twat; Straddleships; straddle shit; Straddle some cows; straddle tale; Straddle the Bench; straddle the fence; straddle the gauze; Straddlewhipped; straddle worth |
| Zero-Shot | Define straddle. straddle synonyms, straddle pronunciation, straddle translation, English dictionary definition of straddle. v. straddled , straddling , straddles v. tr. 1. a. To stand or sit with a leg on each side of; bestride: straddle a horse. b. |

| Query | stonewalling definition |
|---|---|
| Human | Definition: When a spouse is stonewalling in communication in a marriage relationship, he or she is usually.1 using delaying or stalling tactics, or. 2 refusing to answer questions, or. 3 doing whatever can be done to hinder or obstruct a discussi |
| **Oracle** | Stonewalling is a refusal to communicate or cooperate. Such behaviour occurs in situations such as marriage guidance counseling, diplomatic negotiations, politics and legal cases. Body language may indicate and reinforce this by avoiding contact and |
| BM25 | stonewall riot. Stonewall Riot definition. A disturbance that grew out of a police raid on the Stonewall Inn, a popular hangout for gays in Manhattans Greenwich Village in 1969. Such raids long had been routine, but this one provoked a riot as the cr |
| Zero-Shot | Examples of stonewall in a Sentence. 1 They stonewalled until they could come up with a response. 2 They were just stonewalling for time. 3 Theyre trying to stonewall the media. 4 Were trying to get the information, but were being stonewalled. |

*Table C1.* Examples of final human evaluation tasks on MSMARCO dataset (Part 1 of 2). The answers retrieved from the highlighted methods were chosen as the correct labels.

| Query | how many calories do i burn to lose weight |
|---|---|
| Human | How many calories a day do you need to burn to lose weight? A: Calorie Secrets states that 1 pound of fat is equal to 3,500 calories. Therefore, in order to lose 1 pound of body fat you must burn 3,500 more calories th... Full Answer |
| Oracle | How many calories do I need to burn to lose just one pound? 3500 calories: You have to cut down 3500 calories in your diet to loose one pound if you cut down 500 calories per day, you will loose 1 pound in on week, and approx 4 lbs in a month. ...Rea |
| **BM25** | A 3500 Calorie Deficit Approximately 1 Pound of Weight Loss. If you are trying to lose weight through calorie counting, you will need to know how many calories it will take to lose a pound of weight. Creating a calorie deficit of about 3500 calories |
| Zero-Shot | So the amount of energy you exert in doing an activity is measured by the calories burn rate. How to burn calories? Thats easy, just be alive! Your body is constantly burning calories to keep your body functioning. To burn more calories, do more acti |

| Query | how did the uluru rock form |
|---|---|
| Human | UluruAyers Rock is a rock in the Northern Territory in Australia. It is made of red sandstone. It is not the largest rock in the world, being second to Australias Mt Augustu s, which is almost twice the size.yers Rock, now known by its original name |
| Oracle | Uluru, formerly known as Ayers Rock, is a large rock located in the Northern Territory. Ayers Rock was named after the 19th century Premier of South Australia, Sir Henry Ayers.It is located in UluruKata Tjuta National Park, 350 kilometres southwest o |
| BM25 | When did Uluru become a national park? In 1950 Ayers Rock, today known as Uluru, was declared a national park. In 1958 both Ayers Rock and Mt Olga (Kata Tjuta) were excised from an Aboriginal reserve to form the Ayers Rock Mt Olga National Park. |
| **Zero-Shot** | Uluru is easily the most iconic natural landform in Australia, and its formation was equally special. The creation of Uluru and Kata Tjuta as both were formed at the same time began over 500 million years ago.At this time the big crustal blocks tha |

| Query | does quotation mark go before or after period |
|---|---|
| Human | 1 When the whole sentence except for the section enclosed in quotation marks is a question or exclamation, the question or exclamation mark goes outside the quotation mark. 2 When only the unit in quotation marks is a question or exclamation, the ma |
| Oracle | Proper placement of the period with quotation marks. If a sentence ends with quoted material, the period is placed inside the closing quotation mark, even if the period is not part of the original quotation. Note, however, that if the quoted material |
| BM25 | after the quotation marks because if put before the quotation mark, that makes the quote seem like if it continues after what you wrote even if the quote has ended. period mar ks go before the quotation mark because that is ending a sentence... peri |
| Zero-Shot | Do period marks come after parenthesis or before? In a sentence like this, does the period mark come before or after parenthesis? I walked to the door (but I didnt see it was closed). or I walked to the door (but I didnt see it was closed.) What abou |
| **None of the above** (Ground Truth 1) | If the quote is the complete sentence in itself, then the period goes inside the quotation mark. If the quote is part of a larger sentence, then the period goes after the quotation mark. Here is an example: I spoke to her and she told me I don't like |

*Table C2.* Examples of the final human evaluation tasks on MSMARCO dataset (Part 2 of 2). The answers retrieved from the highlighted methods were chosen as the correct labels. For completeness, we also include examples where human selected "None of the above" answer, where we reveal the ground-truth labels in the original dataset. See Appendix C for further discussions.

| Query | when does the new season of law and order svu come on |
|---|---|
| **Human** | Law & Order: Special Victims Unit (season 4): Filming for Season 4 began while Season 3 was still airing as evidenced by reports that Sharon Lawrence would appear on SVU in time for May sweeps.[1][2] |
| Oracle | Law & Order: Special Victims Unit (season 19): Michael Chernuchin, who had previously worked on Law & Order, Law & Order: Criminal Intent, and Chicago Justice took over from Rick Eid as showrunner. This is also the first season since season twelve in |
| BM25 | Law & Order: Special Victims Unit: Executive producer Neal Baer left Law & Order: SVU as showrunner at the end of season twelve, after eleven years (seasons 212) on the show, in order to sign a threeyear deal with CBS Studios.[11] Baer was replaced b |
| Zero-Shot | Law & Order: Special Victims Unit: In 2016, a New York Times study of the 50 TV shows with the most Facebook Likes found that SVUs popularity was atypical: generally slightly more popular in rural areas and the Black Belt, but largely restricted to t |

| Query | who won the peloponnesian war and how did they win |
|---|---|
| **Human** | Peloponnesian War: Sparta and its allies, with the exception of Corinth, were almost exclusively landbased powers, able to summon large land armies which were very nearly unbeatable (thanks to the legendary Spartan forces). The Athenian Empire, altho |
| Oracle | History of the Peloponnesian War: The History of the Peloponnesian War (Greek: , Histories) is a historical account of the Peloponnesian War (431404 BC), which was fought between the Peloponnesian League (led by Sparta) and the Delian League (led by |
| BM25 | Melos and the Peloponnesian War: [4] When looking to find examples of realism, there is a definite bias that comes into play. This is one that arises from a desire to prove realism is an always evident paradigm that can explain past and future occurr |
| Zero-Shot | History of the Peloponnesian War: For the most part, the History does not discuss topics such as the art and architecture of Greece. |

| Query | when did the three little pigs come out |
|---|---|
| Human | The True Story of the 3 Little Pigs!: The True Story of the 3 Little Pigs! is a childrens book by Jon Scieszka and Lane Smith. Released in a number of editions since its first release by Harper & Row Publishers in 1989, and republished the name of Vi |
| **Oracle** | The Three Little Pigs: The Three Little Pigs was included in The Nursery Rhymes of England (London and New York, c.1886), by James HalliwellPhillipps.[1] The story in its arguably bestknown form appeared in English Fairy Tales by Joseph Jacobs, first |
| BM25 | The Three Little Pigs: The third little pig builds a house of bricks. The wolf fails to blow down the house. He then attempts to trick the pig out of the house by asking to meet him at various places, but he is outwitted each time. Finally, the wolf |
| Zero-Shot | The True Story of the 3 Little Pigs!: This is the story of the 3 little pigs from the perspective of Alexander T. Wolf. The wolf is trying to set the story straight of how he came to be big and bad. At the beginning of the book, he is cooking a cake |

| Query | when does the movie the star come out |
|---|---|
| Human | The Star (2017 film): The first trailer was released on July 26, 2017.[20] On November 16, 2017, the official video for the song The Star, performed by Mariah Carey, was made available on her YouTube channel.[21] |
| **Oracle** | The Star (2017 film): In July 2016, the release date was set for November 10, 2017,[18] but it was later pushed back to November 17, 2017.[19] |
| BM25 | Monsters University: Leonard Maltin of IndieWire praised the animation and art direction, but wrote that he wished the movie was funnier and wasnt so plotheavy and that Pixar has raised the bar for animated features so high that when they turn out a |
| Zero-Shot | Movie star: Movie stars in other regions too have their own star value. For instance, in Asian film industries, many movies often run on the weight of the stars crowd pulling power more than any other intrinsic aspect of film making. |

*Table C3.* Examples of final human evaluation tasks on Natural Questions dataset (Part 1 of 2). The answers retrieved from the highlighted methods were chosen as the correct labels.

| Query | who is the longest serving manager in manchester united history |
|---|---|
| Human | Alex Ferguson: Ferguson was appointed manager of Manchester United in November 1986. During his 26 years with Manchester United he won 38 trophies, including 13 Premier League titles, five FA Cups, and two UEFA Champions League titles.[9] He was knig |
| Oracle | Premier League: The leagues longestserving manager was Alex Ferguson, who was in charge of Manchester United from November 1986 until his retirement at the end of the 201213 season, meaning that he was manager for all of the first 21 seasons of the P |
| **BM25** | 201213 Manchester United F.C. season: On 8 May 2013, Uniteds long time manager, Sir Alex Ferguson announced that he would retire from his position as manager of Manchester United after 26 and a half years in charge, making him the longestserving mana |
| Zero-Shot | Ryan Giggs: The son of rugby union, and Wales international rugby league footballer Danny Wilson, Giggs was born in Cardiff but moved to Manchester at the age of six when his father joined Swinton RLFC. Predominantly a left winger, he began his caree |

| Query | when does the new season of lost in space come out |
|---|---|
| Human | Lost in Space (2018 TV series): In October 2014, it was announced that Legendary Television was developing a new reboot of Lost in Space and had hired Dracula Untold screenwriting duo Matt Sazama and Burk Sharpless to pen the new series.[12] In Novem |
| Oracle | Lost in Space (2018 TV series): Lost in Space is an American science fiction television series based on a reimagining of the 1965 series of the same name (itself a reimagining of the 1812 novel The Swiss Family Robinson), following the adventures of |
| BM25 | Lost in Space: In early 1968, while the final thirdseason episode Junkyard in Space was in production, the cast and crew were informally made to believe the series would return for a fourth season. Allen had ordered new scripts for the coming season. |
| **Zero-Shot** | Lost in Space (2018 TV series): Toby Stephens, speaking about the distinction between the original series and the new show: |

| | who plays rachel on jessie punch dumped love |
|---|---|
| Human | Julia Garner: Garner has also acted in another Netflix series, Maniac, as Ellie. Ellie is the sister of main character Annie, played by Emma Stone.[8] |
| Oracle | Jennifer Veal: Jennifer Anne Veal (born 7 September 1991) is a British actress and comedian from Coventry, England. She is best known for her work on YouTube formerly alongside Lucas Cruikshank, as well as her role as Agatha on the American televisi |
| BM25 | Adam Sandler: In 2013, he guest starred in the Disney Channel Original Series Jessie as himself. He and Cameron Boyce previously worked together in Grown Ups and Grown Ups 2. The episode is titled Punched Dumped Love. Sandler costarred in the drama f |
| Zero-Shot | List of Saved by the Bell characters: From sophomore year until the end of the series, Jessie dates athlete A.C. Slater in an opposites attract relationship, which causes friction between the both of them. Slaters pet name for Jessie is Mama. Jessie |
| **None of the above** (Ground Truth 1) | List of Jessie episodes: Guest stars: Lenny Jacobson as Ted the Delivery Guy, Isabella Palmieri as Rachel Kapowski, Jackson Odell as Gale |

*Table C4.* Examples of the final human evaluation tasks on Natural Questions dataset (Part 2 of 2). The answers retrieved from the highlighted methods were chosen as the correct labels. For completeness, we also include examples where human selected "None of the above" answer, where we reveal the ground-truth labels in the original dataset. See Appendix C for further discussions.