# OpenReview forum: "Active Learning with Crowd Sourcing Improves Information Retrieval"
_ICML.cc/2023/Workshop/ILHF — ILHF Workshop ICML 2023_

### Official Review · Reviewer_wk7j · 2023-06-16
**The paper proposes an approach for tuning cold-start search from human preferences. Experiments provide some evidence of effectiveness, though the paper would benefit from better justification and discussion of design decisions.**

**Rating:** 6
**Confidence:** 4

**Review:**

This paper tackles active learning to get human feedback for tuning cold-start search. The CCR approach uses an ensemble of unsupervised models to learn query responses. Then, top answers from members of the ensemble are presented to humans on random queries for preferences. Multiple labelers are used with Dawid-Skene voting to remove noise and incentivize high quality labels. The models are tuned with the human responses using an MNRL loss. Experimental results show CCR improves retrieval performance on two datasets.

Pros:
- Approach shows knowledge transfer from human fine-tuning to unseen queries.
- David-Skene calibration provides interesting way to improve label quality.
- Approach effective with human data and evaluation.

Cons:
- Despite being labeled as an active learning approach, the approach randomly selects items to query before evaluating them with a fixed ensemble to present options to labelers. This seems to lose information that could be gained by actively selecting items to query based on the information they will provide. Since there is an ensemble of responses for each query, the entropy of this distribution could be used as an acquisition function to optimize information gain of preferences (https://arxiv.org/abs/1910.04365), which might improve performance. The authors should discuss these connections and limitations.
- Engineering decisions / ensemble design should be more thoroughly justified (ablations would help).
- A figure showing how the ensemble is used and queries are selected would greatly help understanding.

---

### Official Review · Reviewer_sLvm · 2023-06-18
**Good paper that could be presented better**

**Rating:** 7
**Confidence:** 4

**Review:**

This paper is about actively collecting human feedback (in the form of choice queries where users select one out of multiple options) for cold-start search, which is a problem that I was not very familiar with but the paper gives several examples. The paper proposes to use an EM-like optimization method to do active learning for fine-tuning a model that has been warm-started with unsupervised data.

The paper already contains a great amount of work that is in the standards of a conference paper, but it could definitely benefit from better presentation.

For example, as it can be understood from my vague description of the problem and the solution above, the problem setting is difficult to understand, and it would be useful to have some concrete examples (with actual data). For example, what is the data that the unsupervised learning has been done over? What options are presented to humans for them to select the top option?

I am also not sure (or skeptical) about the use of multiple negative ranking loss (Equation 8), as it introduces some assumptions if I am not mistaken. Mathematically does it assume that the user chose the preferred option over all the options in the batch rather than the options presented to the user? What’s the justification for that? Perhaps it is the fact that the presented options are expected to be better than the other options, but this is an assumption that has not been discussed or justified. Ablation studies about it could improve the paper quality. For example, the current loss could be compared with the vanilla choice model in which the user chooses the preferred option from the presented slate rather than the whole batch.

Another concern that I have is that the paper is not about "implicit" human feedback, because the feedback is actually rather explicit.

All in all, this paper contains interesting application ideas, and I would accept it for the workshop unless this concern of human feedback being explicit is a deal-breaker.

---

### Decision · Program_Chairs · 2023-06-20

Accept